# Evolution and Adaptation of the Avian H7N9 Virus into the Human Host

**DOI:** 10.3390/microorganisms8050778

**Published:** 2020-05-21

**Authors:** Andrew T. Bisset, Gerard F. Hoyne

**Affiliations:** 1School of Health Sciences, University of Notre Dame Australia, Fremantle WA 6160, Australia; gerard.hoyne@nd.edu.au; 2Institute for Health Research, University of Notre Dame Australia, Fremantle WA 6160, Australia; 3Centre for Cell Therapy and Regenerative Medicine, School of Biomedical Sciences, The University of Western Australia, Nedlands WA 6009, Australia; 4School of Medical and Health Sciences, Edith Cowan University, Joondalup WA 6027, Australia

**Keywords:** H7N9, avian influenza virus, hemagglutinin, neuraminidase, polymerase basic protein 2, evolution, mutation, reassortment

## Abstract

Influenza viruses arise from animal reservoirs, and have the potential to cause pandemics. In 2013, low pathogenic novel avian influenza A(H7N9) viruses emerged in China, resulting from the reassortment of avian-origin viruses. Following evolutionary changes, highly pathogenic strains of avian influenza A(H7N9) viruses emerged in late 2016. Changes in pathogenicity and virulence of H7N9 viruses have been linked to potential mutations in the viral glycoproteins hemagglutinin (HA) and neuraminidase (NA), as well as the viral polymerase basic protein 2 (PB2). Recognizing that effective viral transmission of the influenza A virus (IAV) between humans requires efficient attachment to the upper respiratory tract and replication through the viral polymerase complex, experimental evidence demonstrates the potential H7N9 has for increased binding affinity and replication, following specific amino acid substitutions in HA and PB2. Additionally, the deletion of extended amino acid sequences in the NA stalk length was shown to produce a significant increase in pathogenicity in mice. Research shows that significant changes in transmissibility, pathogenicity and virulence are possible after one or a few amino acid substitutions. This review aims to summarise key findings from that research. To date, all strains of H7N9 viruses remain restricted to avian reservoirs, with no evidence of sustained human-to-human transmission, although mutations in specific viral proteins reveal the efficacy with which these viruses could evolve into a highly virulent and infectious, human-to-human transmitted virus.

## 1. Introduction

The pandemic potential of the influenza A virus (IAV) is well known, with the most significant impact occurring during the 1918 Spanish Flu, where mortality was estimated between 21.5 million and 100 million [1]. In the one hundred years since this initial event, evolutionary adaptations in human and animal influenza viruses have resulted in another three IAV pandemic events; the 1957 Asian flu (H2N2), the 1968 Hong Kong flu (H3N2) and the 2009 swine flu (H1N1) [2]. While pandemic events remain limited in number, recurring seasonal influenza virus epidemics result in approximately three to five million cases of severe illness annually, with between 290,000 and 650,000 deaths linked to virally associated respiratory diseases [3]. The morbidity rate for influenza epidemics underscores the constant molecular changes taking place within the viral genome, which in turn facilitates the evasion of host immunity. In response to selective evolutionary pressures, the IAV is adapting, resulting in viral diversity and the creation of novel genotypes. The emergence of the novel IAV H7N9 in 2013 and the resulting morbidity and mortality signalled an evolutionary adaptation of unknown consequence. The purpose of this review is to document the emergence of the H7N9 virus, how it adapted to human hosts, and also highlight the molecular changes that could bring about a human-to-human pandemic.

## 2. Viral Characterization and Origin of Avain Influenza A(H7N9) Viruses

Influenza viruses are enveloped negative-sense, single-stranded RNA (ssRNA) comprising a segmented genome (Figure 1) [4,5,6]. The three largest RNA segments (1–3) encode the viral polymerases PB1, PB2 and PA, which are necessary for RNA synthesis and replication within an infected cell. Two RNA segments (4 and 6) encode the viral glycoproteins hemagglutinin (HA) and neuraminidase (NA), respectively, covering the virion surface at a ratio of approximately 4:1 [7]. The HA protein mediates binding and viral entry via specificity for host cell surface sialic acid (SA) residues, which are common to many animal species and cell types, whilst NA acts to cleave terminal SA residues, facilitating viral release [7]. Nucleoprotein (NP) is encoded on Segment 5, and mainly serves to bind the segmented RNA genome. The viral RNA Segment 7 encodes proteins that enclose the virion to provide a structural scaffold (M1) and a proton ion channel required for viral entry and exit (M2) [6,7]. The non-structural protein 1 (NS1) and nuclear export protein (NEP) are encoded by RNA Segment 8. NS1 has a major role in restricting the host cell immune response by limiting interferon production, as well as modulating viral RNA replication, viral protein synthesis and host-cell physiology [8]. NEP mediates the export of viral RNA from the nucleus to the cell cytoplasm [9].

The segmented nature of the IAV genome enables genetic reassortment, a process by which complete viral segments are exchanged within a cell co-infected with differing influenza viruses. Reassortment of an IAV genome can then generate an antigenic shift, in which the resulting virus may produce novel antigenic proteins for which there is no pre-existing immunity. The mixing of viral genomes also enhances viral diversity, since strains from different animal species may mix freely within a susceptible host. In addition, selective environmental pressures can facilitate rapid viral evolution through processes like adaptation to a new host environment, evasion of the host immune response or acquisition of antiviral drug resistance [10].

The poor proof-reading capacity of the influenza virus RNA polymerase contributes to molecular changes within the viral genome via amino acid substitutions, deletions or insertions [11]. These changes are responsible for generating viral diversity, and are referred to as antigenic drift [12]. Point mutations result in relatively minor changes at antigenic sites on target proteins, although these changes can accumulate over time, and eventually produce a strain that is no longer recognized by host antibodies [7]. As will be discussed below, antigenic shifts can produce a whole new viral strain, although in the case of the H7N9 virus, antigenic drift can significantly alter the pathogenicity and virulence in humans. 

Wild aquatic bird species are recognized as a major reservoir for influenza viruses, including IAV, providing viral seeding for domestic birds and mammals [12,13]. While IAVs circulate widely in aquatic birds, they also circulate in humans, pigs, horses, domestic birds (including chickens, turkeys, ducks and geese), dogs, marine mammals and wild migratory birds (Figure 2) [10,14,15,16,17,18,19]. Although aquatic birds are recognized as the primary source of IAVs, the circulation of two additional subtypes (H17N10 and H18N11) is carried by bats [13,20], and although these subtypes are phylogenetically similar to IAV, they cannot reassort with IAV [6]. However, a recent study isolated an IAV in Egyptian bats with viral characteristics indicative of an avian host origin. It was experimentally verified that viral replication of this IAV was possible in the lungs of infected mice, thereby demonstrating evidence of a capacity to infect other mammalian species [21]. Although IAVs from bats are not considered potential IAV reservoirs, this recent evidence suggests novel subtype viruses may be emerging from this host.

Figure 2 outlines the diversity of hosts in which replication and mixing (reassortment) of the IAV RNA genome can occur [13]. Wild aquatic birds remain the principal reservoirs for IAVs and all 16 HA subtypes [12,13,22], with the reassortment of IAVs potentially occurring in wild aquatic birds, poultry, swine and humans. Importantly, it should be emphasized that while swine are not considered principal reservoirs for IAV, the potential for influenza viruses normally circulating in three distinct species (humans, swine and birds) to meet within pigs indicates they could act as mixing vessels, providing an ideal host for reassortment and cross-species transmission of novel IAVs [10].

Identification of the avian influenza A(H7N9) virus was first reported in March 2013, when three Chinese nationals were hospitalized with a severe lower respiratory tract disease of unknown cause [23]. Two patients from Shanghai (identified with strains A/Shanghai/1/2013 and A/Shanghai/2/2013) died within six days of hospital admission, while the third from Anhui Province, east of Shanghai (A/Anhui/1/2013), died 19 days after admission. This event was the first sign of a newly emerging virus that had the potential to be severely pathogenic within the human population [23].

Early analysis of H7N9 viruses revealed an unusually high internal genetic diversity, noting the unusual characteristic that gene segments encoding the viral HA and NA genes were more conserved than the segments encoding internal genes [24]. The precise origin of H7N9 viruses is unknown, with internal genes potentially derived from avian H9N2 viruses (A/brambling/Beijing/16/2012), while genes encoding the viral HA and NA were obtained from unknown avian H7N?/H?N9 viruses of Eurasian origin [25,26,27]. Phylogenetic analysis of multiple H7N9 viruses supports a minimum two-step sequential reassortment for generating avian influenza A(H7N9) viruses (Figure 3). This analysis proposed at least two stages of sequential reassortment, incorporating distinct H9N2 viruses at each stage. A lack of data on the first reassortment in wild birds precludes an accurate dating of that event, although the latest reassortment potentially occurred in early 2012 [25].

## 3. Hemagglutinin Mutations Confer Specificity for Human Epithelial Cells of the Respiratory Tract

The viral life cycle is initiated through attachment to a susceptible host. To achieve attachment, a virus must have binding specificity for certain surface molecules on a host cell (a receptor), and without that specificity, transmission of viral particles into the host will not occur. Sialic acid (SA) is a nonspecific term used for nine-carbon acidic amino sugars that act as a cell surface receptor determinant for all influenza A virus (IAV) strains [28]. The attachment of all IAV strains to cells requires SAs with an affinity that are dependent upon the presence of an α-2,3 or α-2,6 linkage between the SA and the sugar galactose. Regarding the α-2,3 linkage, the carbon at Position 2 on the SA hexose is linked via an oxygen atom to the carbon at Position 3 of galactose. This is also the case for α-2,6 except the SA carbon is now linked to Position 6 on the hexose of galactose. On the surface of an IAV, hemagglutinin (HA) glycoproteins are responsible for binding with SAs on the host cell surface, although the specificity of HA towards SA varies between different animal species and contributes to host range restriction. For example, an avian IAV will have an HA-glycan binding specificity that promotes transmission between bird species, but lacks the necessary binding specificity to readily spread to humans, and vice versa [29]. The HA-glycan receptor interaction is critical for human infection [23]. Specifically, it requires an α-2,6 SA residue on the human host [30]. Hence, a human IAV will preferentially bind to an α-2,6 SA residue, while an avian IAV has specificity for an α-2,3 SA residue, thereby limiting the effective transmission of IAVs between birds and humans.

Following the emergence in 2013 of low pathogenic avian infleunza A(H7N9) viruses, evidence began to mount of a novel IAV originating from an avian reservoir, but without signs of sustained human-to-human transmission. The switching of viral HA specificity from avian α-2,3 SAs to human α-2,6 SA receptors confers an increased binding affinity in the upper respiratory tract of humans [23,31,32,33]. It should be noted that epithelial cells within the nasal mucosa of the upper respiratory tract of humans are dominated by α-2,6 linked SAs, while alveolar epithelial cells in the lower respiratory tract are dominated by α-2,3 linked SAs [30]. Efficient replication of an avian IAV in the lower respiratory tract is possible where the avian receptor is present, although replication in the upper respiratory tract is required for sustained human-to-human transmission. Although susceptible individuals are at risk from multiple modes of transmission, aerosol transmission is possibly the predominant mode of IAV transmission [34]. Hence, to achieve effective airborne transmission of avian influenza viruses, there needs to be aerosol transmission via coughing/sneezing followed by adherence to a host cell receptor [34,35]. Switching of the viral H7N9 HA specificity to favour human SA receptors is likely to have been facilitated by mutation, and it has been shown that a change in the viral HA molecule is a crucial adaptation in the transmissibility of previous pandemic influenza strains [36]. Importantly, it has been shown that of the five waves of H7N9 viruses, there is no evidence for enhanced or sustained human-to-human transmission [37].

With reference to avain H7N9 viruses, a change in the SA receptor affinity from avian to human has been consistently linked to the substitution of glutamine (Q) with leucine (L) at Position 226 of HA (HA-Q226L) [23,31,32]. The presence of a leucine residue in Position 226 of HA was predicted to increase the strength of binding affinity of HA to human α-2,6 linked SA receptors [23], yet the H7N9 HA failed to show an increased binding affinity. Instead, the H7N9 HA protein displayed limited binding to α-2,6 linked SA receptors in the upper respiratory tract [33]. It was subsequently demonstrated that mutation of the glycan receptor binding site of H7N9 HA with G228S produced extensive binding to human tracheal tissue (Figure 4). 

An experimental comparison of the glycan receptor binding site for A/Anhui/1/2013 with H3 HA (its phylogenetically closest human-adapted HA) has highlighted the critical importance of S228 in H3 for an amino acid network containing residues S186, T187 and E190 (Figure 4). This network structurally positions E190 such that it can make critical contact with sialic acids for both avian and human receptors. The H7 HA does not possess this amino acid network, as it has G228, which alters the positioning of the contact residues involving E190, and as a result, binding to human respiratory epithelial cells is diminished. Site-directed mutagenesis was used to introduce the G228S mutation into H7 HA to produce HA-G228S, modifying the structural network and optimally positioning E190 and S228 for binding to both avian and human SA receptors [33].

As previously discussed, mutations in HA-226Q are indicative of the H7N9 mammalian adaptation process, with typical human isolates displaying a characteristic HA-Q226L mutation [31,32,38]. While HA-226L continues to provide evidence of preferential binding to α-2,6 SAs in humans, this substitution does not necessarily impart significant binding avidity [39]. Despite this, experimental results demonstrate that HA-Q226L remains critical for binding to α-2,6 SAs, and enables the transmission of H7N9 viruses in mammalian hosts [38]. Should H7N9 viruses with the HA-226L mutation move into the swine population, this could represent a significant selective advantage, given that swine are recognized as mixing vessels for human, swine and avian influenza viruses [40]. Using a recombinant wild-type virus (rAnhui-WT), Liu et al. [38] demonstrated the effectiveness of direct contact transmission amongst pigs using the recombinant Anhui-HA-Q226L mutation [38]. This study identified that transmission/replication-enhancing mutations were occurring after a single passage in pigs, concluding that the potential for novel reassortments to occur with other IAVs was significant, should H7N9 become enzootic within pigs.

Despite the mutation at HA-Q226L, H7N9 retains stronger specificity for avian type SA receptors [41]. Sustained transmission in humans is postulated to require additional amino acid substitutions with specificity for α-2,6 linked SA residues [31]. All human pandemic IAV strains have specificity for α-2,6 linked SA receptors [42,43], which is consistent with the switch from avian to human SA receptors that are accompanied by mutations within the HA receptor-binding pocket. In a systematic mutational analysis, de Vries et al. [31] investigated the effect additional mutations would have towards establishing complete human-type receptor binding specificity [31]. The authors used site-directed mutagenesis of the wild type influenza A virus (A/Shanghai/2/2013), where HA-Q226L was already present. Two substitutions, each containing three amino acid mutations (V186G/K-K193T-G228S or V186N-N224K-G228S), successfully demonstrated an acute loss of binding to α-2,3 SA receptors whilst increasing avidity for α-2,6 SA receptors and subsequent effective binding to human trachea epithelial cells, as shown in Figure 5. This approach successfully demonstrated how a combination of amino acid mutations in H7 HA could result in increased specificity of the H7N9 virus to the α-2,6 linked SA receptors on epithelial cells. The authors concluded that such a mutation would raise the potential for human-to-human viral transmission via airborne droplets, which could lead to a pandemic outbreak [31].

Following the initial outbreak in 2013, the H7N9 virus has undergone reassortment and mutation, resulting in five epidemic waves of infection, with each wave responding to new evolutionary pressures and adapting in the process [44]. The first four waves caused by H7N9 strains were classified as low pathogenic avian influenza (LPAI), although with the emergence of the fifth wave during late 2016, the LPAI H7N9 virus mutated into a highly pathogenic avian influenza virus [45]. The geographic distribution of H7N9 was more widespread in the fifth wave, which was coupled with an increase in human infection clusters, a trait symptomatic of an actively evolving virus [39]. 

To assess the potential for a fifth wave IAV switch in receptor specificity, triple mutations previously identified by de Vries et al. [31] (V186G/K-K193T-G228S) were introduced in two separate lineages of the H7N9 virus isolated from the 2016 fifth wave (Yangtze River Delta and Pearl River Delta), and investigated to assess changes in human SA receptor specificity [39]. The expression of new variant HAs from the recombinant viruses was analyzed using glycan microarray and bio-layer interferometry to show a significant, but incomplete loss of binding to α-2,3 SA receptors, accompanied by increased binding to α-2,6 SA receptors. The incomplete loss of α-2,3 SA receptor affinity in a fifth wave virus contrasts with the complete loss of receptor affinity for similar mutations in A/Shanghai/2/2013, implying a subtle mutagenic shift in this latest wave [39]. Yang et al. [39] noted that while HA receptor-binding preference is important, it is not the only consideration for efficient human-to-human transmission [39].

The first step in the viral lifecycle is attachment to a susceptible host epithelial cell, without which infection cannot ensue, and the life cycle terminates. Epithelial cells of the upper respiratory system express abundant α-2,6 SAs at their surface, while alveolar cells in the lower respiratory tract are coated with α-2,3 SAs. The adaptation of H7N9 into the human host from an avian reservoir is realized because HA mutations have switched their binding affinity from α-2,3 SAs in birds to α-2,6 SAs in mammals, and consequently can bind to epithelial cells in the upper respiratory tract. A consistent substitution of leucine at HA-226 occurs in H7N9 viruses, but this mutation does not endow the expected strong binding, potentially contributing to the lack of sustained human-to-human transmission. In contrast to this, a single point mutation at 228 (G228S) produced a significant binding affinity to human cells, and served to highlight the ease with which this virus could mutate into a highly infectious form.

## 4. Mutations in Polymerase Basic Protein 2 Enhance Replication and Virulence of H7N9

Successful binding to the human host is a necessary first step in viral transmission, followed by host cell entry and replication. As a part of the viral replication complex, polymerase basic protein 2 (PB2) plays a crucial role in mammalian adaption, hence an amino acid change or reassortment in PB2 has the potential to allow more efficient viral replication in a new host. Replication efficiency in mammalian cells can be linked to amino acid substitutions in PB2 [36,46], therefore like hemagglutinin (HA) mutations, a change in virulence factors may be related to the accumulation of specific PB2 protein mutations.

Avian influenza viruses typically carry glutamic acid (E) at Residue 627 in PB2. Replication efficiency and host specificity are known to be influenced by Residue 627 in PB2. The substitution PB2-E627K has been linked to increased pathogenicity in human isolates of H5N1 and H7N7, both highly pathogenic avian influenza viruses [47]. Thus, there is some evidence that 627K may be instrumental in enhancing the replication efficiency and virulence of avian influenza viruses in mammals, and potentially contribute to mammalian adaptation [48]. 

Mok et al. [49] identified increased polymerase activity associated with PB2-E627K in human isolates of H7N9, noting that poultry isolates lacked this mutation. The effect of substituting lysine for glutamic acid (PB2-E627K) was experimentally investigated using mice infected with viruses encoding either the PB2-627E or PB2-E627K proteins. Results showed a decrease in disease severity, lower virus replication and decreased pro-inflammatory cytokines in mice lungs following viral infection with the influenza A virus (IAV) (PB2-627E), compared to mice infected with IAV (PB2-627K), concluding that these mutations contribute to increased pathogenicity in mice and mammalian adaptation [49]. Furthermore, in addition to PB2-E627K, human isolates of H7N9 viruses have been identified with additional PB2 mutations, including Q591K and D701N, which are all known to contribute to mammalian adaptation [49,50,51]. Similar substitutions in avian H7N9 viruses have not been reported, noting that although these mutations occur rapidly in mammalian isolates of H7N9, they are mammalian-specific, and are likely to have occurred after infection from the avian host [49]. Yamayoshi et al. [51] provided experimental evidence for increased viral polymerase activity associated with mutations PB2-627K and PB2-701N, concluding that these mutations are essential for the mammalian adaptation of H7N9. 

Xiao et al. [52] demonstrated how PB2-A588V resulted in enhanced polymerase activity, viral replication and virulence for avian-origin H7N9 viruses in mammalian and avian cells. It was noted that the presence of this mutation (PB2-A588V) has increased in human-origin H7N9 viruses, from 0% in 2013 to 24.2% in 2014, with only a minimal presence of this mutation in H7N9 viruses of avian origin (1.9%) [52]. It was concluded that PB2-588V is essential for mammalian adaptation, and when coupled with PB2-627K, significantly affects the replication and virulence of H7N9.

An increase in the replication efficiency of avian influenza viruses in humans is associated with specific mutations in PB2. The similar occurrence of a single amino acid substitution (PB2-E627K) in H7N9 and other highly pathogenic viruses suggests this substitution is instrumental in mammalian adaptation. Other PB2 mutations are known to contribute to replication, enhanced polymerase activity and virulence, and it appears that a virus with multiple PB2 mutations could be appreciably more virulent.

## 5. Neuraminidase Stalk Truncation Enhances Pathogenicity and Virulence of H7N9

Host tropism of the influenza virus is strongly influenced by virus-receptor specificity and avidity for hemagglutinin (HA), preferentially binding to α-2,6 SA receptors, resulting in the fusion of the viral envelope with host cells, whilst neuraminidase (NA) acts to cleave sialic acid (SA) from glycans, thereby contributing to the release of viruses from the cell surface [53,54]. Importantly, the balance between HA and NA activity is considered critical for effective influenza A virus transmission and replication [54,55], hence mutations in the NA stalk of the H7N9 virus may produce changes in virulence factors.

When IAV H7N9 emerged in 2013, a notable characteristic of this virus was the deletion of amino acids 69 to 73 in the NA stalk, a feature consistent with other influenza subtypes. Chen et al. [56] postulated that this was a potential mechanism for increasing human tropism and virulence, although this conclusion was reached without experimental evidence, instead noting that a decreased stalk length was statistically significant in other avian influenza subtypes (H5N1, H6N1, H7N1, H7N3 and H9N2), and therefore may have similar significance in H7N9. Although the shortened NA stalk is considered a remnant of molecular evolution, following the early adaption of IAV from wild aquatic birds to terrestrial poultry, it was notable that this was the first time such a deletion had been observed in N9 [57].

A subsequent study by Bi et al. [57] investigated the impact NA stalk length variation has within the H7N9 virus, specifically if the deletion of the five amino acid sequence (69–73) impacted virus infectivity or replication. H7N9 strains with NA stalk length variations (deletions or insertions) were administered to mice via intranasal inoculation to demonstrate that the five-amino acid deletion (NA 69–73), commonly present in H7N9, had no significant impact on viral replication, NA activity or pathogenesis [57]. It was contended that NA stalk length is optimized as an evolutionary strategy to maintain a functional balance of HA-NA interaction, thereby enhancing viral fitness, and that short deletions have no discernible impact on pathogenicity [57,58]. 

In contrast, certain NA stalk deletions are known to produce virulence enhancements in H5N1 [57,59,60], and therefore similar amino acid deletions (NA residues 49–68, 54–72 and 54–73) were tested in mice by deleting these sequences in three separate mutations (A/Anhui/1/2013) [57]. Truncated NA stalks resulted in significantly greater pathogenic infections in mice, compared with that of a full-length NA stalk virus, although the naturally occurring 5 amino acid deletion in the NA stalk of H7N9 had no significant impact on NA activity, viral replication or pathogenesis in mice [57]. The increase in pathogenicity was attributed to a disruption in the HA-NA balance and the consequence of a shortened NA stalk length, although the link between reduced NA stalk length and increased virulence requires further investigation.

The association between NA stalk length, pathogenicity and virulence in H7N9 has received relatively limited attention within the research literature. Although present in other influenza subtypes, the natural deletion of five amino acid sequences in N9 does not appear to impact viral fitness or pathogenesis. However, expanded sequence deletions of amino acids in NA results in major pathogenic infections, yet the likelihood of this occurring as a natural mutational advantage seems unlikely, since it has been demonstrated that viral fitness is intimately tied to a functional HA-NA balance.

## 6. Conclusions

Evolutionary pressures have driven molecular changes within the viral genome of H7N9, and in the process, established favourable binding to human epithelial cells through increased specificity for α-2,6 linked SA receptors. In an apparent incongruity, the H7N9 virus also retains its specificity for avian cells, which may be an indication of insufficient or ineffective evolutionary pressures. Experimental studies show how a single amino acid substitution enhances binding to human epithelial cells (HA-Q226L), but an equivalent reduction in specificity for avian α-2,3 linked SAs is lacking, and nor is there a particularly strong avidity for α-2,6 linked SAs in the presence of this mutation. The potential impact of single point mutations in hemagglutinin (HA) was recognized when serine was experimentally substituted for glutamine at the HA-228 residue. HA-G228S produced a significant increase in binding affinity within human tracheal cells, signalling the relative simplicity with which antigenic drift could create a highly infectious virus. Introducing multiple mutations into HA residues appears to increase virulence and strengthen binding to human-type cells.

Despite the current lack of specificity by H7N9 HA for human cells, there is enough evidence to show increased virulence, and viral replication occurs under certain PB2 amino acid mutations. The substitution of lysine for glutamic acid at Residue 627 (PB2-E627K) occurs across many highly pathogenic avian influenza viruses, delivering enhanced replication efficiency and virulence in mammals. Like the introduction of multiple mutations in HA, the introduction of multiple PB2 mutations into the viral genome altered the viral function, creating a shift towards increased polymerase activity and virulence. 

With much of the research focus directed towards HA receptors, little attention has been given to molecular changes in the neuraminidase stalk. Naturally occurring five amino acid deletions in the H7N9 NA stalk are noted to occur in other influenza subtypes, and have been shown to have no significant impact on infectivity, replication or pathogenesis. However, the introduction of extended amino acid sequence deletions in NA stalk length resulted in significant increases in pathogenicity. Nevertheless, it seems that the prospect of such deletions occurring naturally may be limited, if recognition is given to the functional balance that must be maintained between HA and NA activity. 

The H7N9 virus continues to evolve through reassortment and amino acid substitutions. While certain mutations have been shown to elicit high pathogenicity, increased virulence and transmission, the prevalence of these mutations appears to be limited at present. Even so, it has been readily demonstrated that a single mutation of one amino acid is all it takes to create a more adaptive H7N9 virus. Under the right conditions, evolutionary pressures could result in a mutation favouring direct human-to-human transmission, at which point H7N9 viruses have the potential to become an unconstrained epidemic, and reach pandemic status within a very short period.

## Figures and Tables

**Figure 1 microorganisms-08-00778-f001:**
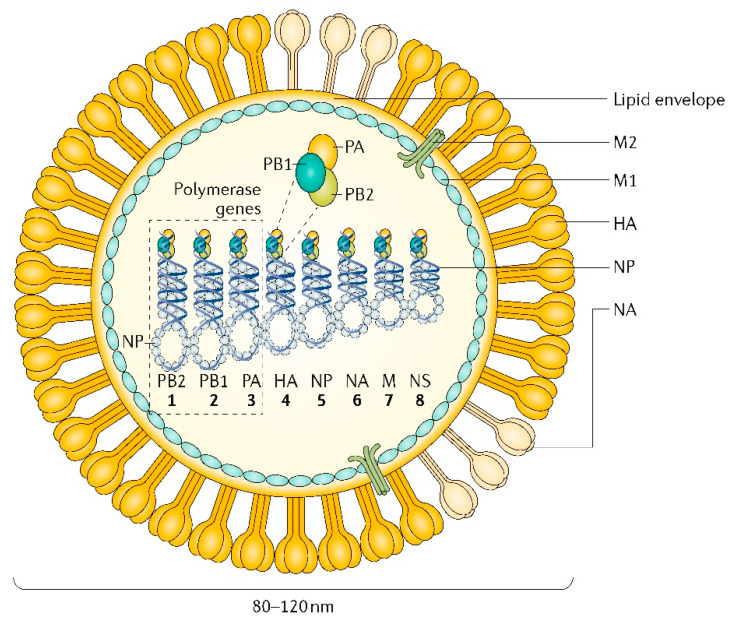
Diagrammatic representation of the influenza A virus (IAV) and its viral genome. Eight internal ssRNA segments encode the major viral proteins of: the RNA-dependent RNA polymerase (PB2, PB1 and PA); HA providing the structural basis for host binding and viral entry; NA facilitating viral release, the binding viral NP; RNA Segment 7 (M) encoding the matrix scaffolding protein (M1) and membrane protein (M2); RNA Segment 8 (NS) encoding a non-structural protein and NEP. Reprinted by permission from Springer Nature: Springer, Nature Reviews Disease Primers [6], Copyright (2018).

**Figure 2 microorganisms-08-00778-f002:**
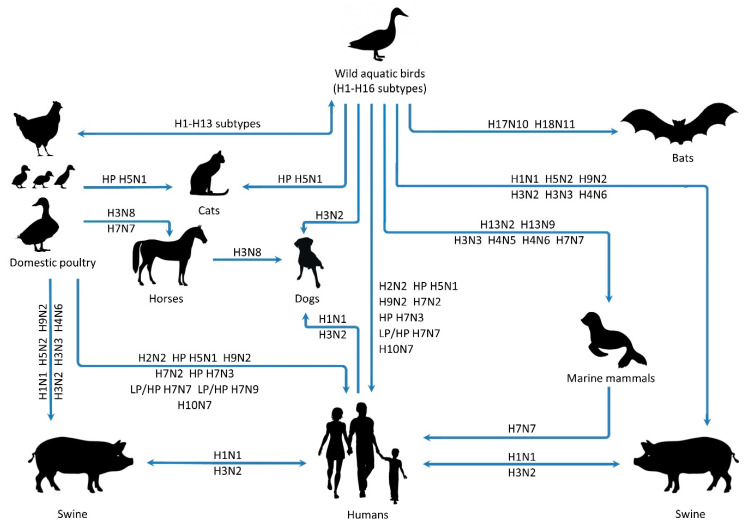
Aquatic birds remain the principal reservoir of all influenza viruses. Cross-species transmission adds to viral reassortment and mixing possibilities with swine acting as mixing vessels for influenza viruses uniquely adapted to birds and humans. Adapted by permission from Springer Nature: Springer, Springer eBook [13], Copyright (2014).

**Figure 3 microorganisms-08-00778-f003:**
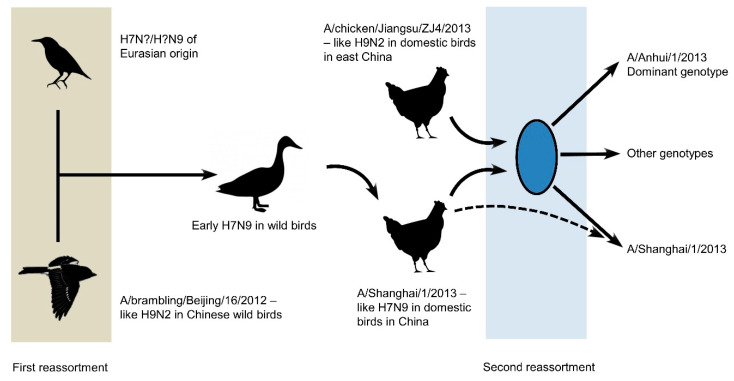
A minimum two-step sequential reassortment was proposed for the evolution of H7N9, with the latest reassortment occurring in early 2012. The first reassortment likely took place between two distinct species of wild birds, incorporating a distinct H9N2 virus. The resulting virus transmitted to Chinese domestic birds before undergoing a second reassortment, with more recent H9N2 viruses already circulating in Chinese poultry. Reprinted/adapted from [25], Copyright 2013, with permission from Elsevier.

**Figure 4 microorganisms-08-00778-f004:**
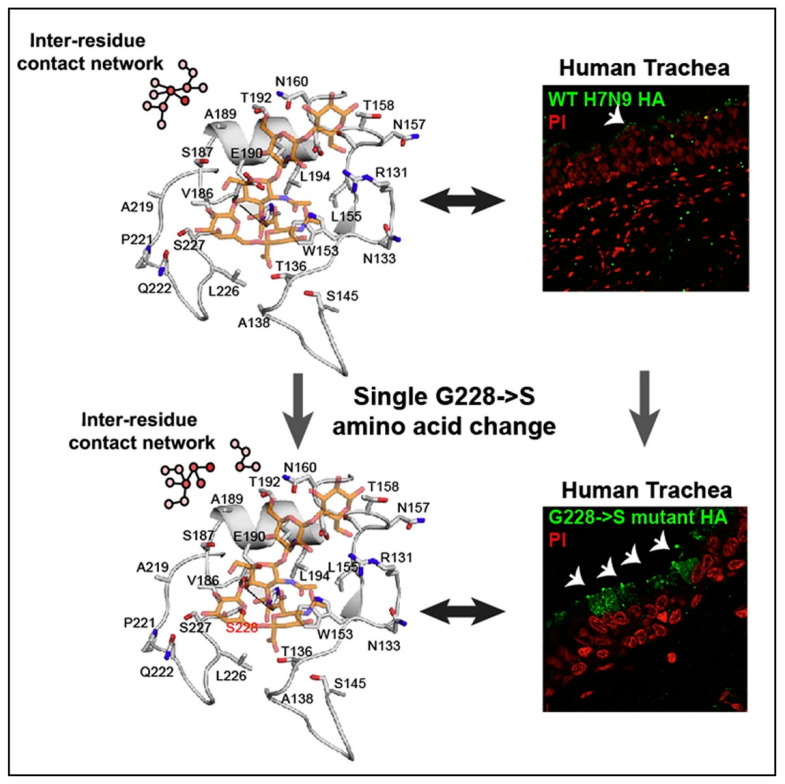
Staining of human trachea cells with G228S hemagglutinin (A/Anhui/1/2013) and a wild type (WT) virus. Tissue sections present recombinant HA stained green, counterstained with propidium iodide in red. The WT virus has not stained cells (top) as intensely as HA-G228S (bottom), demonstrating an increased affinity of the mutated HA for human trachea cells. White arrows indicate specific staining by recombinant HA (in green). Reprinted from [33], Copyright (2013), with permission from Elsevier.

**Figure 5 microorganisms-08-00778-f005:**
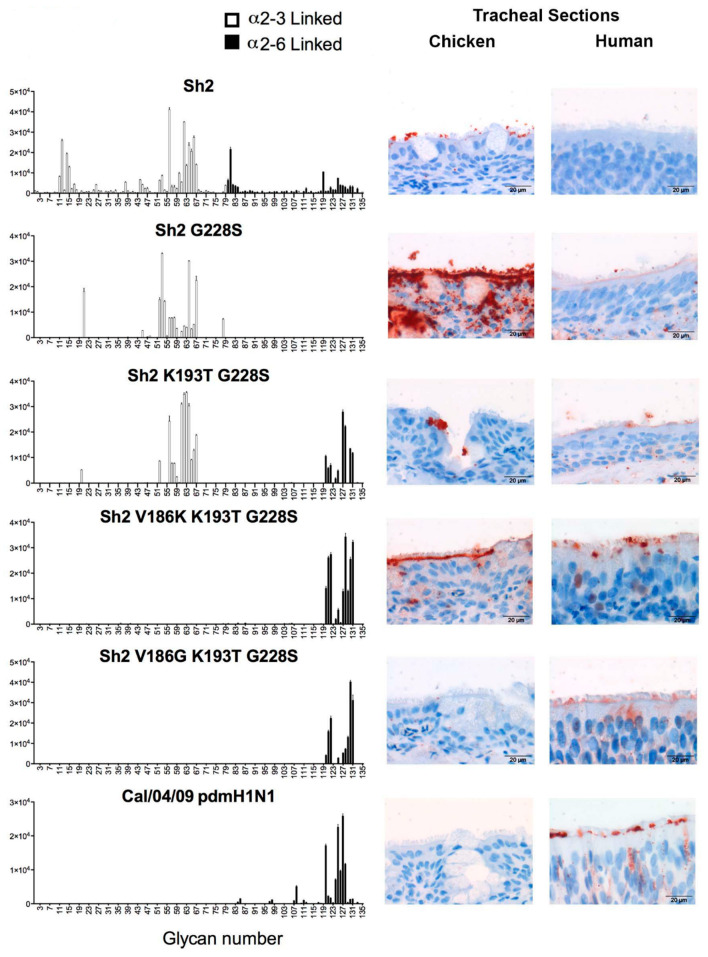
Determination of specificity for wild type (Sh2 = A/Shanghai/2/2013) and mutant H7 HAs on glycan arrays (left) and trachea epithelium (right). Mutations introduced in each glycan array are listed above the plot. Glycans 1 to 10 are non-sialylated controls, while 11 to 79 represent α-2,3 linked sialosides and 80 to 135 represent α-2,6 linked sialosides. The binding profile for triple mutants V186K/G-K193T-G228S is nearly identical to the pandemic control virus Cal/04/09 2009 H1N1. Minimally adapted from the original picture in [31].

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
