# Peer review of "Evolution and Adaptation of the Avian H7N9 Virus into the Human Host"

_microorganisms, 2020, doi:10.3390/microorganisms8050778_

Round 1
Reviewer 1 Report
This manuscript by Bisset and Hoyne is a review focusing on the features of avian H7N9 viruses that were isolated in human cases of infection.
The review contains too many wrong or inaccurate informations, and accumulates disparate fragments of information with no clear and synthetic view. Anybody not familiar with avian influenza viruses would certainly misunderstand and get an inaccurate view of influenza viruses. Further, all the figures are either reprinted or adapted from published articles. This review is not only incomple and inaccurate, but there is no originality at all.
Major remarks (this is not exhaustive).
The title is wrong by several points. The “the Avian H7N9 Virus” does not exist, since there are several H7N9 viruses, even for the viruses isolated from human cases.And there is no “adaptation to man of this virus”. Adaptation would mean that a virus (or a quasi-species) sustainably circulates in human and progressively adapts to human as a result of this circulation. This is not at all the case.
Line 13. What is the meaning of “a low pathogenic reassortment of avian origin viruses”?
Lines 14-17. The sentence seems to mean that a single virus is circulating in humans and adapting to humans. This is wrong.
Line 33. Why 21.5 millions? (20-100 millions is sufficiently precise).
Lines 87-88. “…for influenza viruses, including IAV, providing viral seeding into domestic birds and mammals [12,13]. Whilst IAV circulates”. IAV should be plural, but are there other influenza viruses (other than IAVs) in this reservoir? Line 108. “…[pigs] are susceptible to three distinct IAV species”. What are these distinct IAV species?
Lines 115 and 123. There is no A/China/2013(H7N9) virus. There are instead several H7N9 viruses of avian origin that have been isolated in human cases in China.
Lines 145-47. The authors should explain what are alpha-2,6 and alpha-2,3.
Lines 149-150. The sentence wrongly suggests that the virus could have jumped to human once and for all, then to adapt to humans.
Lines 158-163. It would seem that the authors believe that the H7N9 has gained the ability to transmit between humans (and the following paragraphs would seem to confirm that impression).
Lines 172-73. How can the authors justify that H3 HA is the human-adapted HA that is phylogenetically the closest one to H7?
Lines 164-252. Accumulation of disparate informations, with no synthetic and clear view. It gets more and more obscure.
Lines 242-252. This paragraph somewhat repeats previous paragraphs.
Line 252. “…the ease with which this virus could mutate into a highly pathogenic form.” What exactly do the authors mean by “highly pathogenic form”? Actually, “highly pathogenic” refer only to avian influenza viruses (H5 or H7), which can have either the “High-Pathogenicity” or “Low-Pathogenicity” phenotype (associated with a polybasic cleavage site of the heamagglutinin).
Lines 255-352. Again, accumulation of disparate informations, with no synthetic and clear view. It gets more and more obscure.
Minor remarks
Line 122. The authors could also cite Lam et al. 2013 (PMID: 23965623) (and perhaps also Lam et al. 2015 (PMID: 25762140) for the second H7N9 wave.
Reviewer 2 Report
The review is well written and well illustrated.
I have no criticisms
Reviewer 3 Report
Microorganisms-793257: Evolution and Adaptation of the Avian H7N9 Virus into the Human Host.
In this manuscript, Bisset and Hyone review the changes in H7N9 HA, NA, and PB2 genes and their effects on pathogenicity and transmissibility. The manuscript does a nice job of discussing genes other than HA, as many studies only focus on the HA, and is written clearly with great figures. My comments are minor.
General comment: Several studies have shown that H7N9 viruses can transmit via aerosol droplets using the ferret model. While section 3 is appropriately titled (receptor binding compared to transmission) that these studies do not have to be included, including them would help support some of the paper’s discussion about binding to 2,6 SA receptors.
Specific Comments:
Line 152-reference 23 is not the best reference to solely use here. This paper discussed receptor binding studies in the discussion (but did sequence the H7N9 viruses from the first cases), but did not perform the receptor binding studies in this paper. The original publications performing the studies should be cited in addition to this one. I see that you have them in line 166 (your reference #23 references other papers than the ones you have for 32-34), so they can be added to line 152 as well.
Line 322/323: Please add the appropriate reference for the results.
Round 2
Reviewer 1 Report
In my opinion the authors have not appropriately answered to my remarks, nor have they substantially modified their manuscript.